# miR-489 Confines Uncontrolled Estrogen Signaling through a Negative Feedback Mechanism and Regulates Tamoxifen Resistance in Breast Cancer

**DOI:** 10.3390/ijms23158086

**Published:** 2022-07-22

**Authors:** Mithil Soni, Ozge Saatci, Gourab Gupta, Yogin Patel, Manikanda Raja Keerthi Raja, Jie Li, Xinfeng Liu, Peisheng Xu, Hongjun Wang, Daping Fan, Ozgur Sahin, Hexin Chen

**Affiliations:** 1Department of Biological Science, University of South Carolina, Columbia, SC 29208, USA; mks2232@cumc.columbia.edu (M.S.); guptag@email.sc.edu (G.G.); yogincpatel@gmail.com (Y.P.); keerthim@email.sc.edu (M.R.K.R.); 2Department of Drug Discovery and Biomedical Sciences, South Carolina College of Pharmacy, University of South Carolina, Columbia, SC 29208, USA; ozge.saatci91@gmail.com (O.S.); xup@cop.sc.edu (P.X.); sahinozgur@gmail.com (O.S.); 3Department of Chemistry and Biochemistry, University of South Carolina, Columbia, SC 29201, USA; li439@mailbox.sc.edu; 4Department of Mathematics, University of South Carolina, Columbia, SC 29201, USA; xfliu@math.sc.edu; 5Department of Biomedical Engineering, Stevens Institute of Technology, Hoboken, NJ 07030, USA; hwang2@stevens.edu; 6Department of Cell Biology and Anatomy, University of South Carolina School of Medicine, Columbia, SC 29209, USA; daping.fan@uscmed.sc.edu

**Keywords:** miR-489, breast cancer, estrogen receptor, tamoxifen resistance, CRISPR/Cas9

## Abstract

Approximately 75% of diagnosed breast cancer tumors are estrogen-receptor-positive tumors and are associated with a better prognosis due to response to hormonal therapies. However, around 40% of patients relapse after hormonal therapies. Genomic analysis of gene expression profiles in primary breast cancers and tamoxifen-resistant cell lines suggested the potential role of miR-489 in the regulation of estrogen signaling and development of tamoxifen resistance. Our in vitro analysis showed that loss of miR-489 expression promoted tamoxifen resistance, while overexpression of miR-489 in tamoxifen-resistant cells restored tamoxifen sensitivity. Mechanistically, we found that miR-489 is an estrogen-regulated miRNA that negatively regulates estrogen receptor signaling by using at least the following two mechanisms: (i) modulation of the ER phosphorylation status by inhibiting MAPK and AKT kinase activities; (ii) regulation of nuclear-to-cytosol translocation of estrogen receptor α (ERα) by decreasing p38 expression and consequently ER phosphorylation. In addition, miR-489 can break the positive feed-forward loop between the estrogen-Erα axis and p38 MAPK in breast cancer cells, which is necessary for its function as a transcription factor. Overall, our study unveiled the underlying molecular mechanism by which miR-489 regulates an estrogen signaling pathway through a negative feedback loop and uncovered its role in both the development of and overcoming of tamoxifen resistance in breast cancers.

## 1. Introduction

Oncogenic activation of the estrogen receptor (ER) signaling pathway occurs in over 70% of breast cancers [1]. Although this subtype of breast cancer has the best prognosis due to targeted endocrine therapies, most patients with advanced disease eventually develop resistance to these endocrine therapies. Even for patients treated in an adjuvant setting, a considerable risk of relapse persists indefinitely [2]. Furthermore, approximately 50% of patients with locally advanced or metastatic ER+ breast cancer do not respond to first-line endocrine treatment [3]. Additionally, most patients who initially respond to the therapy eventually develop acquired resistance [4]. Despite significant research efforts and discoveries made in recent years, the exact reasons for endocrine therapy failure in patients with ER+ breast cancer remain largely unknown. Published studies have implicated the mutations in the ESR1 gene, epigenetic silencing of ESR1, activated growth factor receptor signaling, including the EGFR/HER2 pathway, the PI3K-AKT pathway and the MAPK pathway, and overexpression of co-activators such as NCOA3 and FOXA1, as important mechanisms of de novo or acquired resistance [5,6]. However, the only mechanisms of anti-estrogen resistance that are supported in the clinic are HER2 amplification, mutations in the ligand-binding domain (LBD) of ESR, and dysregulation of the CDK4/6 pathway [7,8,9]. Discovery of novel agents that simultaneously modulate these pathways may help the development of improved targeted combination strategies to combat endocrine resistance.

Dysregulation of miRNAs has been increasingly recognized as a critical contributor to cancer development, progression, and therapy resistance. Many miRNAs have been reported to contribute to endocrine resistance. Downregulation of miR-489 has been observed in tamoxifen-resistant breast cancer, but its functional involvement remains unexplored [10,11]. In this study, we systematically investigated the functional roles of miR-489 in ER+ breast cancer. We demonstrated that downregulated miR-489 expression significantly stimulated estrogen-dependent and independent growth and promoted tamoxifen resistance through hyperactivation of p38, PI3K-AKT, and ERK signaling pathways. Thus, patients with ER+ breast cancer, together with low miR-489 expression, may be intrinsically resistant to endocrine therapies.

## 2. Results

### 2.1. miR-489 Expression Is Lost in Tamoxifen Resistance, Predicts Breast Cancer Aggressiveness, and Is Regulated by the Estrogen/ERα Axis

To specifically identify miRNAs that are clinically relevant to endocrine resistance, we analyzed miRNA screening datasets of endocrine-resistant models previously published by three independent laboratories, including our own (Figure 1A) [11,12,13]. Since all three models are established through different processes, they represent independent tamoxifen-resistant models. The MCF7-HER2 cell line acquired resistance through the activation of the HER2 oncogenic pathway, while MCF7-TAM and MCF7:2A represent acquired resistance through long-term culturing in tamoxifen-containing and estrogen-deprived media, respectively. Although many miRNAs were dysregulated in these cell lines, we only found a few miRNAs that were aberrantly expressed in all three cell lines (Figure 1B). Out of these miRNAs, miR-135b, miR-33b, and miR-505 showed an opposite expression pattern among these cell lines. miR-378a-3p and miR-218 were significantly upregulated in all three cell lines, while miR-342-5p and miR-489 were significantly downregulated. Intriguingly, miR-489 was one of the most downregulated miRNAs in all three datasets, suggesting its potential role in tamoxifen resistance. We validated these results using qRT-PCR and indeed found significant downregulation of miR-489 in both resistant cell lines (Figure 1C). To determine whether the expression of miR-489 was associated with endocrine resistance in patient cohorts, we examined miR-489 expression in hormone-therapy-treated ER+ breast cancer patients. We observed a statistically significant association between lower miR-489 expression and poorer overall survival in these patients (Figure 1D). Together, these results suggest that the loss of miR-489 may promote tamoxifen resistance.

Our previous publications showed that the average expression levels of miR-489 were notably higher in luminal cells compared to those in basal cells [12,14]. miR-489 expression was positively associated with ER/PR expression statuses (*p* < 0.0001), but negatively associated with HER2 expression status (*p* = 0.0111) (Appendix A). Furthermore, analysis of miR-489 expression in 13 different breast cancer cell lines demonstrated that it was expressed at a higher level in hormone-positive luminal breast cancer cell lines (Figure 1E) compared to that in cell lines from other subtypes. miR-489 is an intragenic microRNA located in the intron region of calcitonin receptor (CALCR). Analysis of primary breast tumors revealed that miR-489 expression positively correlated with the expression of CALCR, ESR1, and ER-responsive genes such as progesterone receptor (PGR) (Figure 1F), suggesting that miR-489 expression may be regulated by estrogen signaling. To examine this hypothesis, we stimulated three ER+ breast cancer cell lines (T47D, MCF7, and BT474) with estrogen or ethanol for indicated time periods and measured the expression of miR-489 and its host gene, CALCR. We found significant upregulation of miR-489 and CALCR in all three cell lines treated with estrogen (Figure 1G). Estrogen regulation of miR-489 was further investigated in complete media and estrogen-deprived media. As expected, depletion of estrogen drastically reduced the expression of miR-489 and CALCR, similarly to what is observed for other classical ER target genes such as trefoil factor 1 (TFF1), progesterone receptor (PGR), and C-X-C motif chemokine ligand 12 (CXCL12) (Figure 1H). In summary, this data strongly suggests that miR-489 is an estrogen-regulated miRNA in breast cancer and may play a regulatory role in tamoxifen resistance.

### 2.2. miR-489 Overexpression Overcomes Tamoxifen Resistance

Since miR-489 was lost in tamoxifen-resistant tumors and cell lines, we asked whether the restoration of miR-489 would sensitize the resistant cell lines. Our previous studies have shown that overexpression of miR-489 inhibited breast cancer cell proliferation [12,14]. We first tested whether tamoxifen-resistant cells were still sensitive to miR-489 mimics. Notably, we observed that tamoxifen-resistant cell lines, MCF7-TAM and MCF-HER2, were just as sensitive to miR-489 mimics as their tamoxifen-sensitive counterparts (Figure 2A,B). These results raised the possibility that miR-489 might target pathways involved in resistance and could potentially sensitize these resistant cell lines to tamoxifen. Indeed, forced expression of miR-489 significantly sensitized both resistant cell lines to tamoxifen. Tamoxifen alone had no significant effect on the growth of both resistant cell lines at 5 µM, while a combination with miR-489 led to around 40% growth inhibition in MCF7-TAM (Figure 2C) and 30% growth inhibition in MCF7-HER2 cells (Figure 2D). To determine whether miR-489 suppresses cell growth independently or has a synergistic effect with tamoxifen, we performed a synergy analysis. In MCF7-TAM cells, the combination of miR-489 and tamoxifen achieved synergistic effects at high levels (fraction reduction > 0.2). However, in MCF7-HER2 cells, this combination showed a slight synergy (nearly additive effect) at most levels (Appendix A). In both cell lines, this combination could dramatically reduce the dose of either miR-489 or tamoxifen required to achieve the same level of growth inhibition. In the presence of miR-489, the IC50 values of tamoxifen values decreased from 12.1 to 7.8 µM to in MCF7-HER2 cells and from 9.2 to 5.7 µM in MCF7-Tam cells. These results indicated that miR-489 and tamoxifen can synergistically inhibit cell growth in a cell-line-dependent manner.

To further assess the role of miR-489 in the development of tamoxifen resistance, we inhibited endogenous miR-489 in tamoxifen-sensitive cells MCF7-Vec and MCF7-WT. As expected, inhibition of miR-489 significantly increased tamoxifen resistance in both sensitive cell lines. At the highest concentration tested, inhibition of miR-489 increased survival by 25% and 40% in sensitive counterparts of MCF7-Vec and MCF7-WT, respectively (Figure 2E,F). Similarly, colony formation assays also revealed that forced expression of miR-489 significantly reduced the colony-forming ability in resistant cell lines, while inhibition of endogenous miR-489 enhanced the survival of sensitive counterparts. (Figure 2G). To rule out the possibility of the off-target effect of the miR-489 inhibitor, we utilized CRISPR/Cas9 gene editing to create a miR-489 knockout cell line (Appendix A). We validated these knockout cells using genotyping and sequencing to ensure the deletion of miR-489. As expected, miR-89 knockout cells exhibited an increased growth rate (Appendix A). Consistently, miR-489 knockout cells also showed significant resistance to tamoxifen, evidenced by MTT-based cell viability and colony formation assays (Figure 2H,I). These results provided direct evidence to support that the loss of miR-489 contributed to the development of tamoxifen resistance.

### 2.3. miR-489 Acts as an Endogenous Negative Feedback Regulator to Balance Estrogen Signaling

To elucidate the underlying mechanisms of miR-489-induced tamoxifen sensitization, we re-examined the gene expression profiles of T47D cells transfected with miR-489 mimics and scrambled RNA [14]. Interestingly, gene expression analysis revealed enrichment of multiple pathways involved in estrogen signaling and tamoxifen resistance (Figure 3A,B). We observed enrichment of the ErbB signaling pathway and several stress-associated pathways, including endoplasmic reticulum (ER) stress and lysosomal pathways. All of these pathways have previously been reported to be involved in tamoxifen resistance [4,9,15,16,17,18,19]. Meanwhile, these results are also in accordance with our previous studies showing the role of miR-489 in HER2 signaling, metabolic stress, and autophagy regulation [12,14,20]. Interestingly, we noticed significant enrichment of estrogen-dependent gene expression and ESR-mediated signaling. Upon further analysis, we found substantial downregulation of estrogen-responsive genes (Figure 3C). This data suggested that estrogen-regulated miR-489 might function as a negative regulator of estrogen signaling. To determine the clinical significance of this data, we applied a miR-489 gene expression signature to a gene expression profile obtained from patient datasets. Consistently with our microarray results, we found a strong inverse correlation between the estrogen-responsive gene signature and the miR-489 signature and between the PI3K-ERBB2 signature and the miR-489 signature (Appendix A). In addition, we noticed that low miR-489 expression in ER+ breast cancer was indicative of worse overall survival (Appendix A), further supporting an essential tumor-suppressive role of miR-489 in ER+ breast cancer.

To examine how miR-489 negatively regulated estrogen signaling, we measured transcriptional activity of estrogen receptors by performing a luciferase reporter assay with a T47D-ERE-Luc reporter cell line. We observed the inhibition of estrogen receptor transcriptional activity upon miR-489 overexpression and increased activity upon inhibition of endogenous miR-489 (Figure 3D). Consistently, gene expression levels of estrogen-responsive genes were further increased upon estrogen stimulation in knock out cells compared to those in wild-type cells (Figure 3E). We then validated the microarray results by performing qRT-PCR analysis on ER+ and ER– cell lines. As expected, miR-489 caused downregulation of estrogen-responsive genes only in ER+ cell lines including T47D, MCF7, and BT474, but did not affect or in some instances increased the expression of these genes in ER– cell lines such as AU565 and HCT116 (Figure 3F,G). These results suggest that miR-489 regulated the expression of these ERα-downstream genes by inhibiting estrogen signaling.

### 2.4. miR-489 Inhibits ER-Induced Cell Proliferation and Cancer Stem Cell Expansion

Our previous studies have shown that miR-489 inhibits the proliferation of all breast cancer cell lines including ERα+ cell lines [12,14]. However, it remains unknown whether miR489-mediated growth inhibition in ER+ breast cancer cells is due to its effects on ER signaling. To investigate this, we examined the effects of miR-489 on estrogen-induced cell proliferation in MCF7 and T47D cells. Both cell lines showed poor proliferation when treated with the vehicle (ethanol), while estrogen treatment enhanced proliferation by more than two-fold and four-fold in MCF7 and T47D cell lines, respectively. Restoration of miR-489 completely abolished estrogen-induced proliferation in both cell lines, while inhibition of endogenous miR-489 further increased estrogen-induced proliferation by more than two-fold in MCF7 and more than three-fold in T47D cell lines (Figure 4A,B). Similar patterns were observed in colony formation assays by modulating miR-489 expression in MCF7 and T47D cells (Figure 4C,D). Forced expression of miR-489 almost completely inhibited estrogen-induced colony formation of both cells. Interestingly, inhibition of endogenous miR-489 drastically enhanced estrogen-mediated colony formation (Figure 4C). 

Estrogen treatment has been previously shown to enhance the population of stem-like cells in ERα cell lines [20]. These so-called cancer stem cells are thought to be responsible for tumor relapse [21]. We hypothesized that miR-489 may inhibit the estrogen-induced population of cancerous stem-like cells by its effect on estrogen signaling. Therefore, we studied the effect of miR-489 on the estrogen-induced cancerous stem cell population using colony formation assays, mammosphere assays, and flow cytometry. Consistently with previous results, estrogen increased the cancerous stem-like cell population (CD44^+^CD24^−^) three-fold in MCF7 cells and 10-fold in T47D cells. Inhibition of endogenous miR-489 further increased the cancerous stem-like cell population more than three-fold in MCF7 cells and more than 11-fold in T47D cells (Figure 4E). Similarly, we observed increased MFE upon estrogen treatment, and miR-489 inhibition not only increased MFE but also increased the mammosphere size. Forced expression of miR-489 almost completely prevented mammosphere formation (Figure 4F). Together, these results suggest that estrogen-regulated miR-489 is a feedback regulator that is able to confine estrogen-induced tumor cell growth and inhibit the population of cancerous stem-like cells.

### 2.5. miR-489 Inhibits Estrogen-Induced Signaling by Targeting p38 and PTPN11

We further sought to elucidate the molecular mechanism responsible for miR-489-mediated inhibition of the estrogen-ERα axis. Multiple mechanisms have been identified for the regulation of estrogen-ERα-mediated gene expression [22,23]. Direct inhibition of ERα or its co-factors, inhibition of kinases that activate ERα, and inhibition of estrogen-induced nuclear localization of ERα have been previously reported to regulate estrogen-induced gene transcription [24]. We first examined if miR-489 exerted its effects by affecting localization of estrogen receptors. Interestingly, forced expression of miR-489 strongly promoted translocalization of estrogen receptors from the nucleus to the cytoplasm in MCF7 and T47D cell lines (Figure 5A,B). In contrast, inhibition of endogenous miR-489 or knockout resulted in increased nuclear localization (Appendix A). Searching through factors which have been reported to regulate localization of estrogen receptors, we found that one of these factors, p38 MAPK, is a potential miR-489 target [24] (Figure 5C). Interestingly, forced expression of miR-489 significantly downregulated total p38 MAPK (Figure 5D). We then performed a 3′UTR assay to examine if p38 MAPK is a direct target of miR-489. Forced expression of miR-489 significantly reduced the luciferase activity of wild-type constructs but did not affect luciferase activity of constructs with a mutant miR-489 binding site (Figure 5E). This result confirms that p38 MAPK is a direct target of miR-489. Next, we tested if the p38 MAPK inhibitor, SB203580, could recapitulate the effect of miR-489 on estrogen receptor localization. We transfected a control siRNA or miR-489 mimic for 72 h or treated with DMSO or 10 μM SB203580 for 24 h in hormone-starved cells, followed by treatment with estrogen to examine estrogen-induced nuclear localization of estrogen receptors. Indeed, T47D cells treated with a p38 MAPK inhibitor recapitulated the effect of miR-489 on ER localization (Appendix A). However, co-treatment of MCF7 cells with a p38 inhibitor following miR-489 transfection could not further decrease ERα phosphorylation or enhance cytoplasmic translocation of ERα (Figure 5F), indicating that miR-489 affects nuclear translocation of ERα at least partially by downregulating p38 MAPK.

Considering that phosphorylation of the ERα protein is also a critical step for its transcription activity, we investigated changes in the phosphorylation status upon modulation of miR-489 expression. Western blot analysis of total ERα protein and its phosphorylated forms suggest that forced expression of miR-489 reduced the phosphorylation of ERα at both S118 and S167 sites (Figure 5G). This data suggests that miR-489 may further regulate ligand-dependent activation of ERα through inhibition of kinases that phosphorylate ERα at residues S118 and S167. MAPK and AKT have been known to regulate ERα phosphorylation at S118 and S167, respectively. Indeed, miR-489 restoration significantly reduced the activated form of these kinases, while its inhibition or knockout enhanced their activation (Figure 5G).

All three kinase pathways have been shown to be regulated by SHP2 and promote an increase in tamoxifen resistance [4,25]. It is also known that the PTPN11 gene encoding SHP2 is the direct target of miR-489 [12,26]. To test whether miR-489 inhibits phosphorylation of these kinases through regulating SHP2, we examined the effect of miR-489 modulation on SHP2 expression and the downstream signaling pathways in MCF7-HER2 and its isogenic cell line, MCF7-Vec, in the presence or absence of the SHP2 inhibitor, RMC4550 (Figure 5H). Western blot analysis confirmed the upregulation of SHP2, pAKT, and pERK in MCF7-HER2 compared to that in MCF7-Vec. Transfection of miR-489 resulted in the downregulation of SHP2 and p38 expression and decreased phosphorylation of AKT and MAPK. SHP2 inhibitor treatment alone inhibited phosphorylation of MAPK and AKT but had no effect on p38 expression or phosphorylation. SHP2 inhibitor treatment of miR-489-transfected cells showed some degree of synergy to inhibit phosphorylation of MAPK and AKT. Overall, our data suggested that miR-489 inhibits phosphorylation of MAPK and AKT at least partially though SHP2, but directly targets p38.

### 2.6. Pharmacological Inhibition of p38 MAPK, PI3K-Akt, and MAPK Recapitulates the Effect of miR-489 in ER+ Breast Cancer Cell Lines

To examine whether pharmacological inhibition of all three responsible kinase signaling pathways recapitulated the effect of miR-489, we inspected estrogen-induced transcription and proliferation after inhibition of all three kinases. To a variable extent, we observed significant inhibition of transcriptional activity of ERα using an ERE-reporter cell line by each kinase inhibitor (Figure 6A). qRT-PCR analysis also showed downregulation of estrogen-responsive genes (Figure 6B). Consistently, these findings were also supported by diminished estrogen-induced proliferation upon inhibition of all three kinases (Figure 6C,D). However, downregulation of endogenous miR-489 or miR-489 knockout was partially able to rescue cells from the growth-inhibitory effect of kinase inhibitors (Figure 6E,F). This data suggests that miR-489 may exert its inhibitory effect on estrogen signaling partially, if not completely, by simultaneously inhibiting p38 MAPK, AKT, and ERK signaling pathways.

Intriguingly, we observed that p38 MAPK inhibition had a significant inhibitory effect only in the presence of estrogen and did not affect estrogen-independent growth (Figure 6G), indicating that p38 MAPK activation may be estrogen-dependent. Therefore, we tested if estrogen activated p38 MAPK and then mediated nuclear translocation. Since estrogen-induced ERα nuclear translocation occurs within 5–30 min [27], we performed a time course of estrogen treatment on MCF7 and T47D cell lines. We observed a sharp increase in phospho-p38 MAPK and its downstream target, phospho-ATF2, upon estrogen treatment in both cell lines (Figure 6H). These results are consistent with previous studies that showed estrogen-mediated activation of p38 MAPK in various tissues [24,28]. This provided evidence of a positive feedback loop between the E2-ERα axis and p38 MAPK in breast cancer cells, such that binding of E2 leads to activation of p38 MAPK, and activation of p38 MAPK leads to nuclear translocation of Erα, which is necessary for its function as a transcription factor (Figure 6I). Since p38 MAPK activation was estrogen-dependent, we suspected that p38 MAPK inhibition might have a pronounced effect in premenopausal women as compared to postmenopausal women. Indeed, we observed a higher p38 MAPK gene signature score in premenopausal luminal patients compared to that in postmenopausal luminal patients (Appendix A). Furthermore, luminal premenopausal patients with a higher tumor grade also showed a higher p38 MAPK gene signature (Appendix A). Additionally, our correlation analysis of miR-489 expression and p38 MAPK signature showed statistically significant inverse correlation in premenopausal patients, but not in postmenopausal patients (Appendix A). More importantly, high p38 MAPK expression in premenopausal patients predicted poor survival more significantly compared to postmenopausal patients (Appendix A). In summary, these results suggest that miR-489 regulates tamoxifen resistance by targeting multiple kinases signaling pathways and therefore could potentially be used as a therapeutic sensitizer to treat resistant patients.

## 3. Discussion

Defining the role of the differentially regulated miRNAs in breast cancer and drug-resistant cells could lead to the development of new diagnostic tools and therapeutic approaches. In the present study, we provided new evidence for the role of miR-489 in breast cancer and development of tamoxifen resistance. We demonstrated that expression of miR-489 was induced by estrogen and strongly correlated with cellular ERα status. The induction of miR-489 mainly occurred at the transcriptional level because the mRNA levels of its host gene, CALCR, were increased in several ER-positive breast cancer cell lines upon estrogen treatment and correlated with miR-489 expression levels in breast cancers. However, the induction of CALCR gene expression was more robust than the induction of miR-489 in the ER+ breast cancer cell lines MCF7 and T47D, indicating that additional regulatory mechanisms may be involved. Apart from regulating the expression of miRNAs at the transcriptional level, there is evidence to suggest that ERα may be able to regulate microRNA processing and maturation. For example, ERα regulates the processing of the primary transcripts of the two miRNA clusters mir-17-92 and mir-106a-363, and therefore, miRNAs that are processed from the same precursor transcript accumulated in different relative amounts [29]. Further studies are warranted to find out the detailed mechanisms of estrogen-regulated miR-489 expression in breast cancer cells.

Our previous studies have demonstrated that miR-489 functions as a tumor-suppressor microRNA in breast cancer [12,14]. Estrogen exposure is generally associated with an increased risk for breast cancer. To better understand this apparent paradox, we further studied the signaling interplay between miR-489 and estrogen. Microarray analysis of the gene expression profiles of miR-489 overexpressing T47D cells indicated enriched ER-regulated signaling pathways. Consistently, in silico data analysis of gene expression in large cohorts of breast cancer samples confirmed the inverse correlation between miR-489 signature and estrogen-responsive genes. Molecular experiments validated that overexpression of miR-489 suppressed ERα signaling pathways. These results led us to hypothesize that induction of miR-489 by estrogen may serve as a fail-safe mechanism to prevent overactivation of ERα downstream signaling pathways in response to high levels of estrogen. The corollary of this hypothesis is that the loss of miR-489 will promote estrogen-induced tumorigenesis or aggressiveness in cancer cells. Indeed, we observed that the inhibition of endogenous miR-489 robustly enhanced estrogen-induced cell proliferation and expansion of cancer stem-like cell populations, which contrasts with the relatively mild effects of overexpression of miR-489 in breast cancer cells.

Previous studies have described a negative feedback loop between ERα and several miRNAs that are induced upon estrogenic stimulation and that downregulate ERα [29,30]. For example, estrogen induced the expression of the miR-1792 family of microRNAs, which can restrict estrogen actions in feedback through different ways. Meanwhile, several microRNAs like miR-18a, miR-19b, and miR-20b can directly downregulate ERα expression and other microRNAs such as miR-20a, miR-17–5p, miR-106a, and miR-20b downregulate the expression of ERα transcriptional co-factor AIB1 [29]. ERα is not a predicted target of miR-489. We did not observe any changes in its expression levels after manipulations of miR489 as expected. However, we found that overexpression of miR-489 drastically suppressed ERα phosphorylation at ser118 and Ser167. Given that AKT and ERK kinases are known to phosphorylate ERα at Ser118 and Ser167, respectively, and enhance its transcriptional activity [9,31,32,33,34,35], one mechanism of miR-489 mediated restriction of estrogen activities is through the inhibition of MAPK and AKT activities. In addition, previous studies have demonstrated that p38 kinase can phosphorylate ERα and affect its cellular localization [16,28,36]. p38 is one of the predicted target genes of miR-489, which was validated by transient transfection of a p38 promoter reporter assay. Consistently, overexpression of miR-489 resulted in translocation of ERα from nucleus to cytoplasm, adding another mechanism of miR-489 restriction of estrogen action.

Interestingly, we also observed a positive feedback loop between E2-ERα signaling and p38 MAPK. Estrogen activates p38 MAPK within a few minutes upon binding to its receptor ERα. This activated p38 MAPK leads to nuclear translocation of ERα, which is essential for its transcriptional activity. Similar observations have also been reported for ERK and AKT in regard to estrogen signaling [36,37]. Like p38 MAPK, estrogen also activates AKT and ERK [38]. Activated ERK and AKT then activate ERα by phosphorylating ERα at Ser1018 and Ser167, respectively [39]. miR-489 regulates this positive feedback loop by inhibiting p38 MAPK, AKT, and ERK activity.

We posit that the physiological function of the negative feedback loop between ERα and the miR-489 would be a fail-safe mechanism to avoid high ERα activity. High ERα activity is potentially dangerous for the cell and is a major risk factor for breast cancer. In addition, miR-489 could serve as a potential prognostic marker in ER+ breast cancer where ER+ patients with low miR-489 may possess hyperactivation of E2-ERα signaling and may potentially represent aggressive cancers. Indeed, clinical analysis of ER+ breast cancer patients suggest that patients with low miR-489 expression have worse survival rates and are more likely to develop drug resistance. Despite a strong correlation between expression of ERα and a favorable response to endocrine therapy, 40–50% of patients with ERα+ breast cancer develop resistance or exhibit de novo resistance, and patients with luminal B and ERα+/PGR− breast cancer exhibit a poor response to tamoxifen [40]. The underlying mechanism appears to be deregulation in estrogen receptor signaling pathways due to crosstalk of growth factor signaling pathways such as PI3K/AKT/mTOR and epidermal growth factor receptor (EGFR) crosstalk with ERα signaling [9,16,41,42,43]. Our observations here indicated that miR-489 could potentially serve as a useful therapy sensitizer to treat tamoxifen-resistant tumors. First, our study along with other reports showed that miR-489 is significantly downregulated in tamoxifen-resistant cell lines. Second, miR-489 directly inhibits AKT and MAPK pathways, which are well-established to promote estrogen-independent growth and tamoxifen resistance. Third, miR-489 directly targets HER2 and its downstream molecules including SHP2 and AKT [12]. HER2 overexpression is known to confer tamoxifen resistance through increased bidirectional ER/HER2 crosstalk [4,16,44,45]. Fourth, p38 MAPK can potentiate the ER in part through increased phosphorylation of ER at Thr311 [24] and enhance ER signaling through co-activator regulation [46]. Increased p38 activity has been associated with breast cancer drug resistance and invasion [16]. Our results suggest that miR-489 directly targets p38 to break the positive feedback loop between ER and p38. Furthermore, autophagy and EMT have been reported to promote tamoxifen resistance. MCF7-TAM cells have been reported to undergo EMT and possesses higher basal autophagy [10,47]. Inhibition of both processes has been previously shown to reverse tamoxifen resistance. Interestingly, miR-489 has previously been reported to reverse EMT through inhibition of Smad3 expression which leads to sensitization of doxorubicin [48]. Previously, we have reported its role in autophagy and chemoresistance [14]. All of this evidence supports a general role of miR-489 in the development of tamoxifen resistance.

In summary, this study revealed a new negative feedback loop between miR-489 and estrogen and demonstrated the potential role of miR-489 in the development of ER+ breast cancer and tamoxifen resistance. miR-489-based therapy may be a useful adjuvant therapy not only in tamoxifen-resistant patients but also in treatment-naïve patients, since apart from inhibiting estrogen signaling, it also blocks the HER2-PI3K-AKT and MAPK pathways. These results contribute to the understanding of the complex regulatory pathways regulating ERα activity and may provide insights needed to develop novel therapies for ER+ breast cancer.

## 4. Materials and Methods

The detailed procedures of cell culture, antibody and immunoblot, flow cytometry, qRT-PCR, cytoplasmic and nuclear fractionation, luciferase reporter assay, microarray, tumorsphere formation assay, colony formation assay, CRISPR/Cas9-mediated genomic editing, and immunochemical staining are described in Appendix A.

### 4.1. Cell Lines and Culture

MCF7, T47D, and HCC1954 were purchased from ATCC in 2013. MDA-MB231, MDA-MB-468, MDA-MB361, Hs578T, ZR-75-1, and BT474 cells were obtained from Dr. Saraswati Sukuma (Johns Hopkins University) in 2008. MCF7 vector and MCF7 HER2 cell lines were kindly provided by Dr. Rachel Schiff (Baylor College of Medicine). MCF7-WT and tamoxifen-resistant MCF7-TAMR cells were established as previously reported [13]. Cells were grown under standard conditions [14,49,50].

### 4.2. Tamoxifen-Sensitization Assays

Tamoxifen-resistant cell lines were treated with control siRNA or miR-489 mimic with or without tamoxifen at indicated concentrations for 72 h. Similarly, tamoxifen-sensitive cell lines were treated with a control siRNA or a miR-489 inhibitor with or without tamoxifen at indicated concentrations for 72 h. MTT-based cell viability assays and colony formation assays were carried out to examine tamoxifen sensitization.

### 4.3. Generation of Knock Out Cells

The CRISPR/Cas9 gene editing method was used to generate the miR-489 knockout cell line as shown in our previous publication [50]. Two guide RNAs flanking pre-miR-489 were designed using a guide RNA designing tool (http://guides.sanjanalab.org/#/, accessed on 26 June 2022). G-block guide RNAs were purchased from IDT technologies. Cas9-GFP plasmid was obtained from Dr. Philip Buckhaults. Gblock guide RNA and Cas9-GFP were co-transfected in T47D cells using a T47D avalanche transfection reagent (ez biosystem). Seventy-two hours post-transfection, GFP+ cells were sorted using a fluorescence-activated cell sorter. Sorted cells were then diluted to single cells and seeded into a 96-well plate. The rest of the cells were seeded onto a 10 cm dish. Colonies grown from individual clones were then expanded and screened for miR-489 deletion using genotyping.

### 4.4. Statistical Analyses

Statistical analyses were conducted with R and GraphPad software packages. Student’s t-test or ANOVA were used for comparison of quantitative data. Gene expression profiles of miR-489, its host gene calcitonin receptor, estrogen receptor alpha, and progesterone receptor were evaluated using a published data set containing 1302 breast cancer patients [51] that were stratified by the mean value of miR-489 expression levels. The linear correlations between miR-489 and CALCR, ER-α, and PGR gene expressions in primary breast cancer tissues were evaluated with the Pearson correlation coefficient analysis. The association between miR-489 expression and the clinicopathologic parameters of the breast cancer patients retrieved from the dataset EGAS00000000122 was evaluated by the Fisher extract or the chi-square test. Values of *p* < 0.05 were considered statistically significant.

miR-489 signature was generated by using the most up- or downregulated genes (−2 < FC < 2, n = 304) upon miR-489 overexpression. Expression of the signature genes in patients was converted into z-scores. To calculate a miR-489 signature score for each patient, the sum of the z-scores of downregulated genes was subtracted from the sum of the z-scores of upregulated genes [14]. PI3K_ERBB2, p38/MAPK, and estradiol-responsive gene signatures (https://reactome.org, accessed on 24 October 2018) in patients were generated by summing up the z-scores of the signature genes for each patient [51]. Significance for the survival analysis was calculated following a log-rank test using published datasets METABRIC and GSE19783.

## Figures and Tables

**Figure 1 ijms-23-08086-f001:**
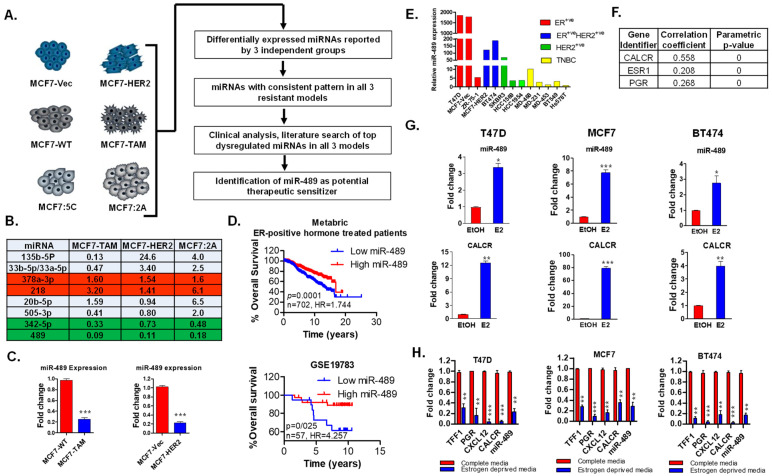
miR-489 expression is lost in tamoxifen resistance, predicts breast cancer aggressiveness, and is regulated by the estrogen/ERα axis. (**A**) Schematic diagram for the identification and validation of miRNAs involved in tamoxifen resistance. miRNA analysis from three independent studies using three independent tamoxifen-resistant model systems. (**B**) List of top dysregulated miRNAs in all three tamoxifen-resistant cell lines. (**C**). qRT-PCR validation of miR-489 in MCF7-TAM and MCF7-HER2 cell lines. (**D**) Clinical analysis of two datasets analyzing miR-489 expression in ER+ breast cancer patients receiving hormone therapy. (**E**) miR-489 expression in breast cancer cell lines. (**F**) Correlation of miR-489 expression with expression of ERα and PGR. (**G**) qRT-PCR analysis of miR-489 expression upon estrogen stimulation in three ER+ breast cancer cell lines. (**H**) qRT-PCR analysis of miR-489 expression upon estrogen deprivation in three ER+ breast cancer cell lines. * *p* < 0.05; ** *p* < 0.01; *** *p* < 0.001. Data are representative of three independent experiments.

**Figure 2 ijms-23-08086-f002:**
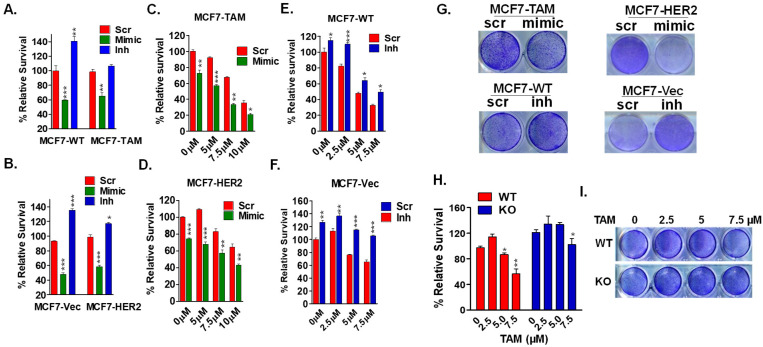
miR-489 restoration overcomes acquired and de novo tamoxifen resistance. (**A**,**B**) Effect of miR-489 modulation on proliferation of two pairs of tamoxifen-resistant cell lines. Cells were transfected with 28 nM of Scramble siRNA, miR-489 mimic, and miR-489 inhibitor for 72 h, followed by an MTT-based viability assay. (**C**,**D**) miR-489 restoration sensitized MCF7-TAM and MCF7-HER2 cell lines to tamoxifen. Scramble siRNA or the miR-489 mimic were transfected with or without tamoxifen for 72 h, followed by an MTT-based viability assay. (**E**,**F**) Depletion of miR-489 promoted tamoxifen resistance in tamoxifen-sensitive cell lines MCF7-TAM and MCF7-HER2. (**G**) Colony formation assay showing that miR-489 modulated tamoxifen resistance. Cells were treated with indicated microRNA mimics or inhibitors with or without tamoxifen for 72 h, followed by a colony formation assay for 7–10 days. (**H**,**I**) miR-489 knockout conferred tamoxifen resistance. WT and KO T47D cells were treated with an indicated concentration of tamoxifen, and viability was examined using MTT (**H**) and colony formation (**I**) assays. * *p* < 0.05; ** *p* < 0.01; *** *p* < 0.001. Data are representative of three independent experiments.

**Figure 3 ijms-23-08086-f003:**
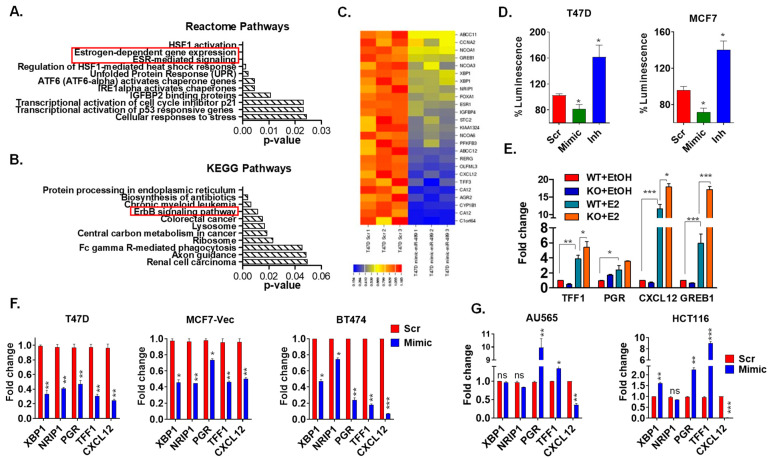
Gene expression analysis revealed enrichment of multiple pathways involved in estrogen signaling and tamoxifen resistance. (**A**,**B**) Whole-transcriptome analysis of miR-489-transfected and control T47D cells followed by pathway enrichment analysis revealed multiple pathways involved in tamoxifen resistance, including ESR-mediated signaling and ErBB2 pathways. (**C**) Heatmap demonstrating downregulation of multiple estrogen-responsive genes in miR-489-transfected T47D cells. (**D**) miR-489 negatively regulated estrogen-induced transcription by transient transfection of cells with the ERE-reporter system. (**E**) miR-489 WT and knockout T47D cells were treated with ethanol or estrogen for 6 h, and expression of estrogen-responsive genes was examined using qRT-PCR. (**F**,**G**) qRT-PCR analysis of estrogen-responsive genes upon miR-489 restoration showed downregulation of estrogen-responsive genes only in ER+ breast cancer cell lines. ns, not significant; * *p* < 0.05; ** *p* < 0.01; *** *p* < 0.001. Data are representative of three independent experiments.

**Figure 4 ijms-23-08086-f004:**
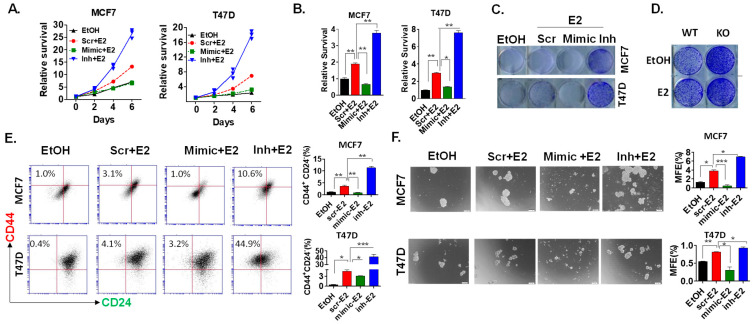
miR-489 acts as an endogenous negative feedback loop to dampen estrogen activities. (**A**) miR-489 restoration inhibited the proliferation of ER+ breast cancer cell lines, while the inhibition of endogenous miR-489 dramatically enhanced the proliferation of MCF7 and T47D. Scramble siRNA, miR-489 mimic, or miR-489 inhibitor were transfected in the presence or absence of estrogen for 72 h, followed by an MTT-based viability assay. (**B**) Quantification of viable cells on day 6 post-treatment of MCF7 and T47D. (**C**) Inhibition of endogenous miR-489 dramatically enhanced estrogen-induced colony formation. MCF7 and T47D cells were transfected with scramble siRNA, miR-489 mimic, and miR-489 inhibitor in the presence or absence of estrogen, followed by a colony formation assay. (**D**) Hormone-starved miR-489 WT and KO T47D cells were seeded in a 12-well plate and treated with ethanol or E2 for 6 days, and cell viability was measured by crystal violet staining. (**E**) Breast cancer cell lines were transfected with scr, mimic, or inhibitor in the presence of estrogen for 72 h, followed by flow cytometry to examine CD24 and CD44 surface markers. (**F**) Breast cell lines were transfected with scr, mimic, or inhibitor in the presence of estrogen for 72 h, followed by a mammosphere assay. Scr = Scramble control; Inh = Inhibitor of miR-489. * *p* < 0.05; ** *p* < 0.01; *** *p* < 0.001. Data are representative of three independent experiments.

**Figure 5 ijms-23-08086-f005:**
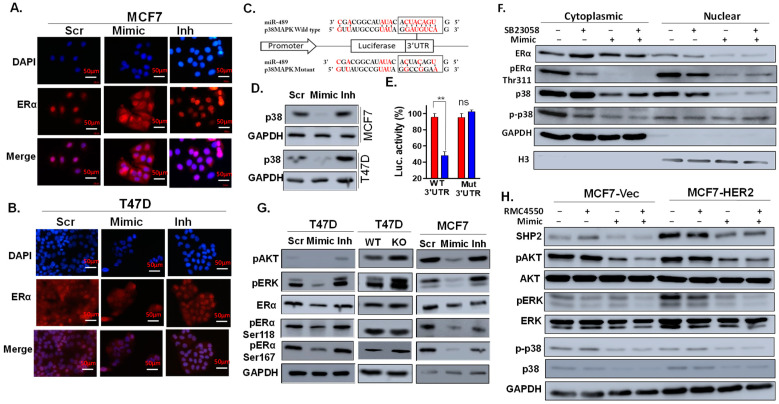
miR-489 inhibits estrogen-induced signaling by inhibiting p38 MAPK, PI3K-AKT, and MAPK-ERK pathways. (**A**,**B**) MCF7 and T47D cells were transfected with a scramble, mimic, or inhibitor for 72 h in estrogen-deprived media, followed by estrogen stimulation for 15 min. The data represent the localization of estrogen receptors using immunofluorescence. (**C**) miR-489 binding site in p38 MAPK 3′ UTR. (**D**) A scramble, mimic, or inhibitor was transfected, and Western blot analysis was performed to examine the total p38 MAPK protein. (**E**) 3′ UTR transfection assay showing that miR-489 directly bound to 3′UTR of p38 MAPK. Data are means of three replicates ± SEM. ns, not significant; ** *p* < 0.01. (**F**) MCF7 cells were transfected with a scramble or mimic in the presence of DMSO or the p38 MAPK inhibitor, SB23508. At 72 h post-transfection, cytoplasmic and nuclear fractionations were prepared for Western blot analysis of expression and phosphorylation statuses of ERα and p38. (**G**) Cells were transfected with a scramble, mimic, or inhibitor for 72 h, and ERα phosphorylation and responsible kinases were examined using Western blot analysis. (**H**) MCF7 cells were transfected with a scramble or mimic in the presence of DMSO or SHP2 allosteric inhibitor RMC4550. ERα phosphorylation and responsible kinases were examined using Western blot analysis.

**Figure 6 ijms-23-08086-f006:**
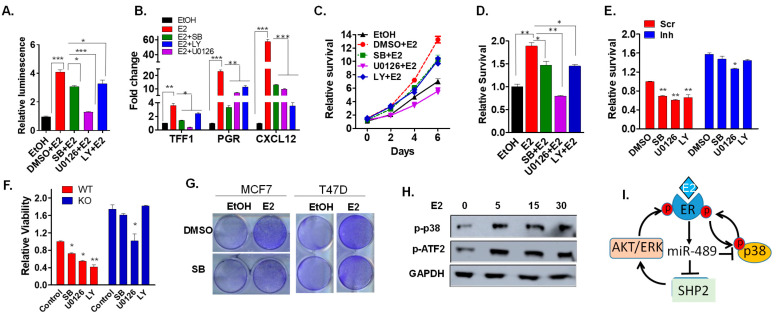
P38 MAPK, PI3K-Akt, and MAPK inhibitors recapitulate the effect of miR-489 in an ER+ breast cancer cell line. (**A**) MCF7 cells were treated with inhibitors of p38 MAPK, ERK, and PI3k-AKT pathways, and their role on estrogen-induced transcription was examined using a luciferase reporter assay. (**B**) MCF7 cells were treated with inhibitors for 24 h, and expression of estrogen-responsive genes was examined using qRT-PCR. (**C**,**D**) MCF7 cells were treated with inhibitors, and the effects on estrogen-induced proliferation were examined using an MTT assay. (**E**) MCF7 cells were transfected with a scramble or inhibitors and treated with inhibitors for 72 h in the presence of estrogen, and the effects on proliferation were examined using an MTT assay. (**F**) miR-489 WT and KO T47D cells were treated with inhibitors in the presence of estrogen, and cell viability was examined using an MTT assay. (**G**) MCF7 cells were treated with DMSO or a p38 MAPK inhibitor in the presence or absence of estrogen and a colony formation assay was performed. (**H**) MCF7 cells were treated with estrogen for the indicated time, and activation of p38 MAPK signaling was examined using Western blot. (**I**) A proposed model for estrogen-induced miR-489 to restrict estrogen signaling via a negative feedback mechanism. * *p* < 0.05; ** *p* < 0.01; *** *p* < 0.001. Data are means of three replicates ± SEM.

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
