# Peer review of "miR-489 Confines Uncontrolled Estrogen Signaling through a Negative Feedback Mechanism and Regulates Tamoxifen Resistance in Breast Cancer"

_ijms, 2022, doi:10.3390/ijms23158086_

Round 1

Reviewer 1 Report

In this manuscript, the authors studied the role of miR-489 in development of tamoxifen resistance. They performed genomic analysis of gene profiles in Tamoxifen-resistant and sensitive breast cancer cells and identified miR-489 as the potential therapeutic target for further mechanistic studies.  A series of well-designed experiments were carried out to mechanistically demonstrate that miR-489 regulates estrogen receptor signaling: i) modulation of ER phosphorylation status by inhibiting MAPK and AKT kinase activities; ii) regulation of nuclear to cytosol translocation of estrogen receptor α (ERα) by decreasing p38 expression and consequently ER phosphorylation. The results shed light on the complex regulatory pathways regulating ERα activity and may lead to development of novel ER+ breast cancer therapies. Overall, the data presented data is of high scientific rigor.  There are a few minor issues that need to be addressed.

1. In Fig.3A, the Reactome pathway analysis showed that p21 signaling pathway is activated by miR-489. That raises a question whether miR-489 directly regulates cell cycle machinery. Does the author have any evidence to show whether p21 can be induced or whether miR-489 causes cell cycle arrest through regulating p21?

2.  In section 2.6. Change the word 'phenocopies' to "recapitulates".

3.  The author can consider moving Fig. 6J-K to the supplementary data section.

Reviewer 2 Report

The manuscript “miR-489 confines uncontrolled estrogen signaling through a negative feedback mechanism and regulates tamoxifen resistance in breast cancer” by Mithil Soni and co-authors to demonstrated that loss of miR-489 expression promotes tamoxifen resistance while overexpression of miR-489 in tamoxifen-resistant cells restored tamoxifen sensitives. Mechanistically, we found that miR-489 is an estrogen regulated miRNA that negatively regulates estrogen receptor signaling by using at least the following two mechanisms: i) modulation of ER phosphorylation status by inhibiting MAPK and AKT kinase activities; ii) regulation of nuclear to cytosol translocation of estrogen receptor α (ERα) by decreasing p38 expression and consequently ER phosphorylation. In addition, miR-489 can break the positive feed-forward loop between estrogen-ERα axis and p38 MAPK in breast cancer cells which is necessary for its function as transcription factor. Overall, this study unveiled the underlying molecular mechanism of miR-489 in estrogen signaling pathway and its implication in development and overcome of tamoxifen resistance in breast cancers. However, there are concerns which must be taken into account before the work can be reconsidered for publication.

1.      Figure 1B: The miR-489 raw data should be provided. How to calculate the miRNA value (Fold)?

2.      Figure 1D: The miR-489 data collected from database should be described in Material and Methods. Do author analysis the correlation between miR-489 and clinical information, such as: age, stage, type, chemoresponse, chemotherapy agent?

3.      Figure 1G: Why did author treat cells with EtOH? A solvent control?

4.      Figures 2A-2D: The IC50 of each cells to tamoxifen should be provided.

5.      Figure 2H: The miR-489 expression should be provided.

6.      Figures 5A and 5B: The scale bar should be added.  

7.      Figures 5D-5H: The miR-489 expression should be provided. The raw data of western blot should be provided.

8.      Figure 6G: Why did author treat cells with DMSO and EtOH? A solvent control?

9.      Figure 6K: Do author analysis the correlation between miR-489 and p38/MAPK score?

Round 2

Reviewer 2 Report

The revised manuscript “miR-489 confines uncontrolled estrogen signaling through a negative feedback mechanism and regulates tamoxifen resistance in breast cancer” have adequately addressed my previous concerns and the paper is now acceptable for publication.